# Generalization of Classification of AlkB Family Alkane Monooxygenases from *Rhodococcus* (*sensu lato*) Group Based on Phylogenetic Analysis and Genomic Context Comparison

**DOI:** 10.3390/ijms26041713

**Published:** 2025-02-17

**Authors:** Kirill V. Petrikov, Anna A. Vetrova, Anastasia A. Ivanova, Olesya I. Sazonova, Irina Yu. Pozdnyakova-Filatova

**Affiliations:** Federal Research Center “Pushchino Scientific Center for Biological Research of the Russian Academy of Sciences”, 142290 Pushchino, Russia; phdvetrova@gmail.com (A.A.V.); mrs.ivanova.a.a@gmail.com (A.A.I.); sazonova_oi@rambler.ru (O.I.S.); irafilatova24@gmail.com (I.Y.P.-F.)

**Keywords:** alkane monooxygenase, alkane hydroxylase, *alkB* gene, rubredoxin, *Rhodococcus*, alkane degradation, bioremediation

## Abstract

Alkane-oxidizing bacteria play a crucial role in the global carbon cycle. *Rhodococcus* species are well-known hydrocarbon degraders, distinguished by the harboring of multiple homologs of AlkB family alkane monooxygenases. Although different types of rhodococcal AlkBs have been described, the overall picture of their diversity remains unclear, leaving gaps in the current classification. We conducted a phylogenetic analysis of all AlkBs identified in *Rhodococcus* (*sensu lato*) and examined the genomic context of the corresponding genes. The sequence clustering was well aligned with genomic neighborhoods, allowing both features to be used as criteria for proposing AlkB types that form distinct phylogenetic groups and have characteristic genomic contexts. Our approach allowed us to revise the classification of previously described AlkBs, identifying eight types on their basis, and to propose three new ones. Alkane monooxygenases whose genes are co-localized with rubredoxin genes can be considered a generalized AlkBR type, the most common among all *Rhodococcus*. In the AlkB0 type, which is a paralog of AlkBR, violations of conservativity in known alkane monooxygenase signature motifs were found. Our findings provide a more consistent classification framework for rhodococcal AlkB that prevents the over-reporting of “novel” types and contributes to a deeper understanding of alkane monooxygenase diversity.

## 1. Introduction

Microorganisms capable of utilizing hydrocarbon substrates as their sole source of carbon and energy are widespread in nature. They have been identified in various environments, including soil [1,2,3], coastal sediments [4,5], seawater [6,7], and rhizospheres [8,9,10]. These microorganisms are even found in extreme environments, such as the Arctic [11,12] and Antarctic [13,14] regions. Aliphatic alkanes are some of the most common hydrocarbons in the natural environment. Alkanes enter ecosystems through a variety of sources. They are produced as metabolites by cyanobacteria [15], form the primary components of waxy substances of plant origin [16,17], and arise during the decomposition of plant residues [18]. Natural seepage of crude oil [19,20] also contributes to alkane release. Human activity significantly increases the presence of alkanes in the environment through accidental oil spills [21,22], routine low-level leaks [23], and oil refinery wastewaters [24], making hydrocarbon pollution a critical environmental issue that underscores the need to study hydrocarbon-oxidizing bacteria to develop effective bioremediation techniques [25,26,27].

From a chemical perspective, alkanes are rather inert substrates, so the key step in their metabolism is the activation of the substrate molecule, which occurs under aerobic conditions through terminal or subterminal hydroxylation to form primary or secondary alcohols, respectively [28]. Therefore, when studying alkane metabolism in bacteria, particular attention is given to the enzymes responsible for substrate hydroxylation: alkane monooxygenases (also known as alkane hydroxylases) [29]. Currently, several families of bacterial alkane monooxygenases are known: copper-containing membrane-bound monooxygenases (CuMMO family) [30,31,32,33], soluble monooxygenases CYP153 from the gem-thiolate cytochrome P450 superfamily [34,35,36], flavin monooxygenases (AlmA [37], LadA [38]), and membrane-bound non-heme diiron monooxygenases of the AlkB family, belonging to the class-III diiron proteins [39]. The latter is one of the most common and best-studied alkane monooxygenases [29,40].

The pathway for *n*-alkane metabolism under aerobic conditions was described in detail in studies of the *alkB*-containing OCT plasmid of *Pseudomonas putida* strain GPo1 (*P. oleovorans* strain TF4-1L = ATCC 29347) [41,42,43]. In addition to alkane monooxygenase itself, two other proteins are essential for the first step, alkane hydroxylation: an electron carrier and a reductase responsible for its reduction (typically, rubredoxin and rubredoxin reductase). During substrate oxidation, a two-electron transfer occurs, consuming one molecule of the reducing equivalent NADH and reducing molecular oxygen to water [29]. Further metabolism involves two more steps of sequential oxidation of the resulting alcohol, to an aldehyde, then to a fatty acid, and a third step: formation of fatty acyl-CoA, which undergoes beta-oxidation [40].

The first *alkB* gene discovered was part of the *alkBFGHJKL* operon, which was located on the OCT plasmid and contained not only the genes for the rubredoxins and reductase required for alkane hydroxylation but also genes for the enzymes involved in the next three steps of alkane metabolism mentioned above: alcohol and aldehyde dehydrogenases and acyl-CoA synthetase [44]. Interestingly, rubredoxin reductase was not part of this operon, although it located in close proximity. This gene organization is very common among members of the phylum *Pseudomonadota* [45], which is not surprising since the operon structure is typical of bacterial degradation genes [46,47,48]. Of greater interest is the fact that the *alkB* genes are found as single entities, without co-localization with any known functionally related genes, e.g., in the genomes of *Pseudomonadota* representatives [49,50].

The genus *Rhodococcus*, a typical hydrocarbon-oxidizing representative of the indigenous microbiomes [51], is particularly noteworthy in this regard: the distinctive feature of certain species is the harboring of multiple chromosomal *alkB* homologs [52]. AlkB genes in *Rhodococcus* can occur either as a single gene or as part of operons. Two types of rhodococcal operons are known: the complete *alkB* cluster, which includes two rubredoxin genes, a rubredoxin reductase gene, and a regulatory protein gene, and a partial cluster that does not contain a reductase gene. It is not currently known how widespread these clusters are, or whether there are other clusters in which *alkB* genes are localized with other genes functionally related to alkane metabolism.

The phylogenetic diversity of AlkB in rhodococci is remarkably high: reports of the discovery of novel types are published regularly [53,54]. However, most of such proposals are based on the description of only a few or even a single homolog, and several are based on only partial sequences, which limits the possibility of their phylogenetic analysis. Without a comprehensive analysis of rhodococcal AlkBs, it is unclear whether such novel sequences are truly representative of a whole new group or whether they are rare, unique singletons, how widespread the different AlkB types are, and what species distribution is characteristic of them. For many types of AlkB, the genomic context of the corresponding genes is unknown, making it impossible to determine whether these genes are part of established operons, elements of novel gene clusters, or single entities. Furthermore, some studies have overlooked the intrageneric variability of rhodococcal AlkBs [55,56], revealing gaps in the current classification. While large-scale analyses of hydrocarbon degradation genes in public databases have been performed [45,57,58], the approaches used provide insufficient resolution at lower taxonomic ranks, making the results unsuitable for detailed classification within genera.

In our work, we address these issues by investigating AlkB diversity within the *Rhodococcus* (*sensu lato*) group. Based on the phylogenetic analysis of their amino acid sequences and the study of the genomic context of the corresponding genes, we refined all the known types of AlkB, proposed three new ones, and provided insights into their species-level distribution. Our findings provide a more consistent classification framework for rhodococcal AlkBs that clarifies the position of previously described monooxygenases and prevents the over-reporting of “novel” types. To the best of our knowledge, this is the first comprehensive analysis of AlkB family proteins that utilizes the entire available dataset of amino acid sequences at the genus level. These findings enhance our understanding of alkane monooxygenase diversity and contribute to a deeper understanding of bacterial alkane degradation systems.

## 2. Results

### 2.1. Defining a Rhodococcus (sensu lato) Group

First of all, we defined the taxonomic boundaries of the group for which the AlkB search was to be performed. There are discrepancies in the scientific community regarding the phylogenetic position of some *Rhodococcus* species, associated with their allocation to other genera: *Rhodococcoides*, *Prescottella*, and *Antrihabitans* [59,60]. Taxonomic classification within *Rhodococcus* (*sensu stricto*) is also complicated, for example, in the case of the very genetically close *R. qingshengii* and *R. erythropolis*, which are hard to separate into species [61]. For these reasons, the same objects may have different taxonomic assignments in different databases (Table 1).

All four of the above-mentioned genera are listed in GenBank as separate genera. The *Rhodococcus fascians* clade was separated from the genus *Rhodococcus* in a recent update. Since the NCBI Taxonomy database is not an authoritative source for nomenclature or classification [62], we examined the taxonomy of *Rhodococcus* with the Genome Taxonomy database (GTDB) [63]. In the GTDB classification, the genera *Prescottella* and *Rhodococcoides* are absent, and all their representatives are assigned to different *Rhodococcus* clusters. The genus *Antrihabitans* is present, but the only available genome of *Antrihabitans cavernicola* is assigned to *Rhodococcus cavernicola*. The taxonomy of *Rhodococcus* in the Type (Strain) Genome Server database [64] is largely consistent with the GTDB. According to the List of Prokaryotic names with Standing in Nomenclature (LPSN), the correct genus name for the taxa *Rhodococcoides*, *Prescottella soli*, *Prescottella subtropica* and *Antrihabitans cavernicola* is *Rhodococcus* [65].

As a result, it was decided to cover the largest number of sequences for further work. We identified a provisional group *Rhodococcus* (*sensu lato*), which included all species classified as *Rhodococcus* in at least one database, namely *Rhodococcus* (*sensu stricto*) spp., *Prescottella* spp., *Rhodococcoides* spp., and *Antrihabitans cavernicola*. Hereafter, the names of taxa are given according to the NCBI Taxonomy.

### 2.2. Phylogenetic Analysis of AlkB Sequences and Examination of the Genomic Context

A BLAST search for AlkB family alkane monooxygenases was performed among representatives of the *Rhodococcus* (*sensu lato*) group (Appendix A). After filtering and removing the outliers (Appendix A), 927 non-redundant translated amino acid sequences of length 349–447 were obtained; the minimum and average amino acid pairwise identities were 45.0% and 61.9%, respectively (Appendix A).

Phylogenetic analysis of the translated amino acid sequences demonstrated the formation of a large number of well-isolated clades with high representation (Figure 1). To identify patterns in AlkB clustering and classify them, we studied the genomic context of the corresponding genes. For the *Rhodococcus* (*sensu lato*) group, chromosomes of all genomes with a “complete” or “chromosomal” assembly level and chromosomes of genomes of type strains were analyzed (excluding “Atypical genomes”, “Metagenome-assembled genomes”, and “Genomes from large multi-isolate projects”). A total of 221 assemblies were considered. All strains with unspecified species designations were identified using the Type (Strain) Genome Server and validated in the GTDB (Appendix A). Based on these results, the following types can be distinguished. To designate the AlkB types, we used sequential numbering based mainly on the designations used in previous works, taking into account the chronological order of the type description.

The AlkB1 type was characterized by the localization of the *alkB* gene in the so-called complete *alkB* cluster. It is an operon that includes the *alkB* gene, two rubredoxin *rubA* genes, and the rubredoxin reductase *rubB* gene (Figure 2). In most of the cases considered, the co-directional gene of the TetR/AraC family regulatory protein *alkU* was located downstream. It can also be part of an operon, as demonstrated for the *Rhodococcus jostii* RHA1 strain [66]. However, in *R. aetherivorans*, *R. indonesiensis*, and *R. ruber*, it is directed convergently. It may be noted that in *Mycobacterium tuberculosis* H37Rv, co-transcription of this gene together with the others was not observed [67]. A total of 110 *alkB1*-containing regions were found. Outside the complete *alkB* cluster, the genomic context is variable.

The genes we assigned to the *alkB2* subtype (a total of 115 regions were examined) were localized in a partial *alkB* cluster that included only two *rubA* and *alkU* genes, in the absence of *rubB* (Figure 2). A distinctive feature of the partial cluster was its localization in a fairly conservative *manA*-*ahcY* region. The downstream region was highly conservative and was preserved in all the genomes examined: after *alkU*, the genes adenosylhomocysteinase *ahcY*, thymidylate kinase, two-component sensor histidine kinase *mtrA*, and *mtrB* were localized. The upstream region contained the genes amino acid permease, cation diffusion transporter *fieF*, mannose-6-phosphate isomerase *manA*, phosphoglucose/phosphomannose isomerase *tobH*, phosphomannomutase/phosphoglucomutase *manB*, and DUF3499 domain-containing protein. The upstream part was more variable: there were variants without transporters, and without both transporter and permease, but the other genes were always present. It can be noted that all the listed genes were co-directed, although there are no data on their operon organization.

In five cases, not a partial but a complete *alkB* cluster was localized in the *manA-ahcY* region, i.e., together with the *rubB* gene. We searched for such regions and found that all corresponding AlkB sequences formed a clade designated as type AlkB1A. The opposite was observed for nine other regions: the partial *alkB* cluster, as with the AlkB2 type, was not localized in the *manA-ahcY* region. The genomic context was variable and did not coincide with any of the variants found in the *alkB1* context. All corresponding proteins, designated AlkB2N, clustered within AlkB1 clusters on the tree and did not form a clade, so it is difficult to identify them as a separate type.

Neither AlkB1 nor AlkB2 types formed monophyletic groups. The AlkB1 type forms two clades, while AlkB2 forms three (if not considering the unique clade of two singleton sequences of AlkB2 from *Antrihabitans cavernicola*). At the same time, one of the AlkB2 clades was formed exclusively by sequences of the genus *Prescottella*, in which, however, AlkB2 was the only homolog. A distinctive feature uniting these two types was the co-localization of the *alkB* gene with the *rubA* genes. The genomic context of all other *alkB* types did not contain *rubA*. Also, no sequences were found in which the *rubB* gene was present in the region, but the *rubA* gene was absent. The only exception was *Rhodococcoides yunnanense* strain G21638-S1: the region included *alkU*, convergently located with *alkB* and *rubB*, and the corresponding protein was clustered together with the AlkB1 type.

A significant exception from the AlkB1-AlkB2 sequence cluster, which includes all cases of *alkB* and *rubA* co-localization, was the sequences we assigned to the AlkB8 type (Figure 1 and Figure 2). For them, only four regions were initially considered, three of which belonged to *Rhodococcoides kroppenstedtii* and one to *Rhodococcoides corynebacterioides*. Conserved determinants of the genomic context, which allowed us to isolate this type, were the alpha/beta fold hydrolase menH and glycosyltransferase *mshA* genes upstream of the region and the downstream ROK family transcriptional regulator. Sequences of this type clustered on a fairly distant branch, which was part of a common cluster with AlkB1 and AlkB2. This branch itself was divided into two clades, and the proteins of the regions we analyzed fell into only one of them. Checking the alignment of all proteins in the branch showed that the key difference between the sequences from the two clusters was the absence of the first 30 amino acids from the N-terminus in the clade that included proteins from regions that we did not analyze. It is likely that these are errors in determining the CDS during annotation: longer AlkB8-type proteins were completely aligned to the translated sequences of the corresponding nucleotide regions, which did not confirm the hypothetical deletion. Analysis of the genomic context of all other sequences localized in this branch confirmed the high conservation of the environment. This allowed us to combine all these high-similarity sequences into one AlkB8 type.

A large distant clade was formed by sequences of the AlkB0 type, the genes of which were localized in the *manA-ahcY* region (22 regions), but all other elements of the *alkB* cluster, *rubA*, *rubB*, and *alkU*, were absent, and the *ahcY* gene was localized immediately downstream from *alkB* (Figure 2). The upstream region in this case was more conservative than for the *alkB2* type: the *manA* and permease genes were always located upstream. Since such a frequent localization of the *alkB* gene in the *manA-ahcY* region was found, we checked those regions in genomes where *alkB1A*, *alkB2*, and *alkB0* were not found. In these cases, *alkU* is always localized between the *manA* and the *ahcY* genes. As in the case of the *alkB2* type, the permease gene or both the permease and the transporter can also be localized between *manA* and *alkU*.

Among the AlkB2 and AlkB0 types, the unique organization of the genomic context in *Antrihabitans cavernicola* was particularly noteworthy. Only one genome had been deposited in GenBank for this species (INSDC and RefSeq versions with slight differences in protein sequences, which explain the presence of two leaves on the phylogenetic tree). The genome of this strain contained the *manA-ahcY* region, in which a partial *alkB* cluster was localized, as for the typical AlkB2 type, but another AlkB gene was localized between the cluster and the *ahcY* gene. The AlkB protein corresponding to the first gene from the operon was clustered together with AlkB2, and the protein corresponding to the second gene with AlkB0. Both sequences were quite distant from the other representatives of the types. Interestingly, we found a similar co-localization of a partial *alkB* cluster with *alkB0* in *Nocardia asteroides* genomes. In this case, the *ahcY* gene was localized downstream, and in the upstream, there was a local rearrangement of the region with *manA* (Figure 3).

Further, we could distinguish the following types of AlkB (Figure 1 and Figure 2). The AlkB3 type was co-localized with the downstream class F sortase gene, and the AlkB4 type with downstream glutamate–tRNA ligase *gltX*. The AlkB5 type was represented in 66 regions and formed a large, distinct clade. In the genomic context, the upstream genes were aminotransferase and cold-shock protein, and downstream was a gene encoding either an AbiEi family antitoxin domain-containing protein or a non-ribosomal peptide synthase. The upstream genomic context of AlkB6 included an ABC transport protein gene. The AlkB7 type gene was located between alpha/beta hydrolase and ATP-binding protein. It was found in representatives of the genus *Rhodococcoides* (seven regions), some *Rhodococcus* species (three regions), and *Rhodococcus* of unknown species (four regions).

One of the AlkBs found in the only completed *Rhodococcus globerulus* genome clustered relatively close to AlkB3, although its genomic context shared only one downstream short hypothetical protein. We additionally analyzed all deposited *Rhodococcus globerulus* genomes (seven accessions) and examined the genomic context of two more proteins (from *Rhodococcus* sp. WS3 and *Rhodococcus* sp. MS16 strains) belonging to this clade. All nine tested regions exhibited a conserved organization, with upstream genes encoding a MerR transcriptional regulator and an iron–sulfur domain-containing protein, and downstream genes encoding a hypothetical protein and a GAF/ANTAR domain-containing protein, confirming that the AlkB9 type can be proposed.

Eight very similar AlkB sequences clustered in a distant clade basal to the AlkB5 clade. We tested all proteins of this clade and all corresponding genomic regions. All strains belonged only to the species *Rhodococcoides kroppenstedtii*, and the genomic context was different. We designated this clade as AlkBK, but according to our approach, we could not distinguish these alkane monooxygenases as a separate type. Another five unique sequences, designated as AlkBX type, clustered separately from all other types and had no common elements of genomic context.

The results of phylogenetic analysis indicate a wide variation in sequence similarity within each type, ranging from very high in AlkB6 (minimum amino acid pairwise identity 94%) to relatively low in heterogeneous clusters such as AlkB1, AlkB2 and AlkB0 (about 60%) (Appendix A). Also, apart from the previously described complete and partial alkB clusters, we found no other cases of localization of *alkB* genes together with known genes functionally related to alkane metabolism.

### 2.3. Distribution of AlkB Types by Species

The distribution of AlkB across *Rhodococcus* (*sensu lato*) species was degenerate: different species could contain one set of AlkB types, but representatives of a single species always carried the same set (Figure 4). Most of the genomes contained one to two homologs of *alkB* genes. Three homologs were found in almost all representatives of the *Rhodococcoides* cluster (seven out of nine). Also, three copies were contained in the genome of *Rhodococcus globerulus*. And the group of closest species *R. erythropolis*, *R. qingshengii*, and *R. baikonurensis* is unique: only its representatives carry four (*erythropolis*) and five (*qingshengii*, *baikonurensis*) copies of *alkB*. Four species had no AlkB gene homologs at all: *Rhodococcus antarcticus*, *R. rhodnii*, *R. spongiicola* and *R. xishaensis*. They were distributed along different branches. The species *R. spongiicola* and *R. xishaensis* formed a basal clade in relation to the *Prescottella* clade, *R. rhodnii* was included in a clade with *R. triatomae*, and *R. antarcticus* was the most distant species among all the *Rhodococcus* (*sensu lato*) species considered, which clustered even further than some outgroups. Unfortunately, each of these four species is represented by only a single genome, which limits the formulation of general conclusions.

The most represented type from the taxonomic point of view was AlkB2 (25 species), followed by AlkB1 (19 species). It should be noted that each species had at least one of these types, with the exception of *Rhodococcus artemisiae*, *Rhodococcoides corynebacterioides*, and *Rhodococcoides kroppenstedtii*. On the other hand, the simultaneous presence of genes of types 1 and 2 was observed only within one clade: “*erythropolis*, *qingshengii*, *baikonurensis*”. The AlkB1 sequences of these species cluster in the most distant AlkB1 clade from all other AlkB1s. The AlkB2N type was found in fairly distant taxa: in the *Rhodococcoides* cluster and in the “*Rhodococcus phenolicus*, *Rhodococcus zopfii*” clade. AlkB1A was present in the genomes of four species that fell into one cluster together with *Rhodococcus spelaei*, containing AlkB2. In all members of the genus *Prescottella*, the only type present was AlkB2.

If the distribution of AlkB1 and AlkB2 by species had a polyphyletic character, then the remaining types were distributed within monophyletic groups. With the exception of the species *Antrihabitans cavernicola*, which is controversial in taxonomic position and unique in the organization of the region, AlkB0 was found only within the clade containing all representatives of the genus *Rhodococcoides* (six species) and three representatives of *Rhodococcus*. All representatives of this clade had the AlkB7 type, except for *Rhodococcoides kroppenstedtii* and *Rhodococcoides corynebacterioides*. At the same time, only these last two phylogenetically closest species had the AlkB8 type, and the conditional AlkBK type in *R. kroppenstedtii*. Both AlkB8 and AlkBK were localized in isolated and remote clusters on the AlkB sequence tree.

Alkane monooxygenases AlkB3, AlkB4, and AlkB6 were found in a single clade with only four species. AlkB4 was found in *Rhodococcus qingshengii*, *R. erythropolis*, *R. baikonurensis*, and *R. globerulus*; AlkB6—in *R. qingshengii*, *R. erythropolis*, and *R. baikonurensis*; AlkB3—only in *R. qingshengii, R. baikonurensis*, and one representative of *R. erythropolis*. The AlkB5 type was present in 15 species that made up a monophyletic group, which included *R. rhodochrous*, *R. aetherivorans*, *R. pyridinivorans*, and *R. ruber*.

### 2.4. AlkB Amino Acid Motifs

The search for amino acid motifs using the MEME tool was performed for the entire obtained set of alkane monooxygenase sequences among representatives of the *Rhodococcus* (*sensu lato*) group, as well as for individual types (Appendix A). The main signature histidine-containing motifs were HE[LM]GH[KR], EHN[RHF]GHH, NY[LVI]EHY[AG], and LQRHSDHHA. All of them had high statistical significance (E-value close to zero) (Appendix A). To our knowledge, this is the first de novo search for motifs performed on a large set of alkane monooxygenase sequences. Comparison of motifs found in individual types showed that the main patterns in them coincide, with the exception of the AlkB0 type (see Discussion).

## 3. Discussion

The existence of several homologous alkane monooxygenases in representatives of the genus *Rhodococcus* was first demonstrated in the work of van Beilen et al. [52]. The same group of authors performed a detailed characterization: the genomic context was described, and the possible role of alkane monooxygenases and rubredoxins in the oxidation of alkanes was assessed using heterologous expression [68]. The authors assigned four types, AlkB1, AlkB2, AlkB3, and AlkB4, which correspond to those proposed by us (Table 2, Figure 5).

In the work by Kim et al. [69], amplicons of *alkB* genes were obtained for several *Rhodococcus* strains. On the basis of phylogenetic analysis, the authors proposed distinguishing three new types of AlkB in addition to the four already described: AlkB5, AlkB6, and AlkB7. Since only partial sequences were provided and the genomes of the studied strains were absent, the genomic context is unavailable. However, based on the phylogenetic clustering of the partial sequences of AlkB6 and AlkB7, we propose revising them as AlkB2 and AlkB1, respectively, and consider the AlkB5 type as indeed new.

Another novel AlkB type was proposed on the basis of partial sequence analysis of the *Rhodococcus* sp. TMP2 strain [70]. According to our classification, it indeed represented a separate type, distinct from the previous five. The authors designated it as AlkB5, but did not take into account the results of Kim et al. [69], so we chronologically designated it as AlkB6. The strain described also contained the AlkB1, AlkB2, AlkB3, and AlkB4 types, i.e., a total of five copies of the *alkB* genes, which is fully consistent with our results for the *Rhodococcus qingshengii* species.

Amouric et al. [71] used the AlkB types from Whyte et al. [68] and Kim et al. [69] to classify *alkB* amplicons obtained for the strains *Rhodococcus* sp. (*ruber*) SP2B and *R. ruber* DSM 43338. Two copies of *alkB* were found in each strain. A genomic context containing two rubA genes and one rubB gene was described for one of these genes. This *alkB* together with its homolog was classified by the authors as AlkB7 type according to the classification of Kim et al. [69] which corresponds to AlkB1 type according to ours. Another pair of homologs was proposed to be classified either as a new type or as AlkB5. Our classification supports the second hypothesis. Thus, the AlkB set in both strains corresponds completely to that of *R. ruber*.

Regarding the alkane monooxygenase found in *Rhodococcus* sp. CH91 [72], the authors proposed classifying it as a novel type. Indeed, phylogenetically, this sequence is quite distant from the previously described proteins (Figure 5). However, based on the similarity of the genomic context, we believe that it can be attributed to the already known AlkB5 type. The genome of the strain contained a second homolog, which belongs to the AlkB2 type. This set of AlkB types is typical for *R. rhodochrous* and a number of closely related species. Analysis of the *Rhodococcus* sp. CH91 genome in TSGS showed that it is probably a separate species, but the closest one is *R. rhodochrous*, which has a similar AlkB set.

Analysis of the genomic context of rhodococcal alkane monooxygenases allowed us to identify the AlkB7 type, which contains the upstream gene alpha/beta hydrolase [53]. The same work showed the presence of single AlkB genes in *Rhodococcus fascians*, localized between the permease and *ahcY* genes, i.e., in the conserved *manA-ahcY* region, as we have shown. We proposed naming this type AlkB0 to distinguish it from the others and to emphasize its localization in this specific region.

As can be seen, in a number of the works described above, alkane monooxygenases were classified as a new type solely based on the clustering of their sequences at a sufficient distance from those described previously. With this approach, for example, the types AlkB6 and AlkB7 proposed in the work of Kim et al. [69] should be considered separate from AlkB1, or AlkB-new from the work of Xiang et al. [72] should be considered separate from AlkB5. However, the results of our phylogenetic analysis show that many types are characterized by significant internal heterogeneity of the AlkB types we propose. Maintaining the above-mentioned approach for classifying all AlkB sequences would lead to splitting the phylogenetic tree into many small clades, creating an enormous number of types. This would only complicate the overall picture of diversity given the lack of functional characterization for most of these types. Our classification framework avoids such unnecessary atomization by taking genomic context as one of the criteria, allowing distant phylogenetic clusters to be considered as belonging to the same AlkB type. Despite obvious limitations, such as the inability to propose separate types for the AlkB2N and AlkBK groups, our approach prevents redundant reporting of finding “novel” types and allows for obvious adjustments to the classification as the number of sequenced genomes grows.

The *alkB* sequences have previously been proposed as species markers [73], although this approach does not always provide the desired taxonomic resolution [74]. Our results suggest that the phylogenetic analysis of alkane monooxygenases may have limited use as an additional marker for species identification.

An interesting case is the possibility of distinguishing the species *Rhodococcus qingshengii* and *Rhodococcus erythropolis*, which is a topic of ongoing discussion [61]. The applicability of molecular genetic analysis of *alkB* genes for identifying representatives of these two species has already been demonstrated [75]. Our results confirm these conclusions. Moreover, species separation can be preliminarily achieved simply by determining the copy number of *alkB* genes.

For six strains, we noticed a discrepancy between the GenBank species assignment and the species-specific sets of AlkB types we found for these strains. We identified these strains using the Type (Strain) Genome Server and checked them in GTDB (Appendix A). Clarified species matched the sets of AlkB types in five cases, and the only representative of *Rhodococcus erythropolis* containing five copies of *alkB* as *Rhodococcus qingshengii* was an exception. We have shown for the first time the existence of species in the genus *Rhodococcus* that do not contain *alkB* homologs, where the absence of this gene may itself serve as an indirect indicator of species affiliation. Unfortunately, most of the taxa considered are represented by only a small number of genomes or even a single genome, which limits the formulation of general conclusions.

The presence of four highly conserved histidine-rich motifs had been described for alkane monooxygenases: HEXXHK, EHXXGHH, NYXEHYG, and LQRHXDHHA [68,76]. Nine histidine residues play a key role in the functioning of the enzyme, since they coordinate two iron atoms in the active site and are necessary for maintaining catalytic activity [77,78]. Eight histidine residues from the first, second and fourth motifs are also signatures for stearoyl-CoA desaturase, xylene monooxygenase, and other members of the membrane fatty acid desaturase (FADS)-like superfamily [79,80,81]. Apart from this histidine signature, no significant sequence similarity was found between members of this family and alkane monooxygenases [82]. So, the third motif NYXEHYG (HYG motif) is considered to be specific for alkane monooxygenases of the AlkB family [68,76]. A number of publications refer to the above-listed motifs as maintaining conservation in all alkane monooxygenases [73,83,84]. However, to our knowledge, in the mentioned works by Smits et al. [76] and Whyte et al. [68], the analysis of histidine motifs of alkane monooxygenases was carried out for a limited set of amplicons, and since then, neither an assessment of the conservation of these motifs on larger sets of sequences nor a de novo search for motifs has been performed.

Our search confirms the presence of the four motifs in all alkane monooxygenases of the *Rhodococcus* (*sensu lato*) group with high confidence. But when comparing motifs among individual types we proposed, a difference was observed in the AlkB0 type: the first and third motifs are HEXXHR and NYXEHY[AG], respectively. In the fourth motif, a small fraction of sequences also shows the replacement of the last alanine by threonine (Figure 6). Most likely, the variant positions of K/R in HEXXH[KR] and G/A in NYXEHYG[AG] do not have a significant effect on the coordination of iron in the active site [82]. But based on alkane monooxygenase histidine motifs, sequence annotations have been performed, and primers have been designed for amplicon sequencing and hybridization [73,85,86]. However, given the deviations from conservation we found, this approach leads to an underestimation of the diversity of AlkB. Interestingly, Táncsics et al. [73] mention the possible absence of *alkB* genes in *R. corynebacterioides*, *R. kroppenstedtii*, and *R. rhodnii*. The latter species, indeed, does not contain *alkB* homologs, but the other two do. This confirms the necessity to adjust the described approach, taking into account our results. Obviously, a full-fledged revision of the signature motifs can only be performed when all alkane monooxygenases are examined, but within the *Rhodococcus* (*sensu lato*) group considered by us, it is necessary to correct the first and second motifs as HEXXH[KR] and NYXEHY[AG].

Our findings contribute to clarifying the evolutionary relationships among rhodococcal AlkB homologs, which is an important enhancement to functional annotation because it allows the inference of orthology or paralogy [87]. For alkane monooxygenase genes, considering genomic context is a crucial extension, as the operon localization of the *alkB* genes clearly exhibits functional relationships with their genomic neighborhoods. Indeed, the genes of AlkB1, AlkB1A, and AlkB2 types, which colocalize with rubredoxin genes, form an almost monophyletic group (with the exception of AlkB8). Given rubredoxin’s biochemical role in alkane hydroxylation, the stability and prevalence of such genes’ organization appear to be evolutionarily advantageous. On the other hand, the genomic context of AlkB2 partially coincides with that of AlkB0, particularly in the *manA-ahcY* region. The presence of both AlkB2 and AlkB0 within the *manA-ahcY* region in *Antrihabitans cavernicola* and *Nocardia asteroides* strongly supports the hypothesis of gene duplication, which allows us to consider AlkB2 and AlkB0 as paralogs.

AlkB1 most likely diverged following the acquisition of the rubredoxin reductase gene as an element of the cluster. This is supported by the existence of the AlkB1A-type complete cluster located in the *manA-ahcY* region. Conversely, the hypothesis that reductase was lost from the complete cluster seems unlikely, as rubredoxin co-localization is common across multiple phyla, whereas the presence of the reductase in the cluster is rather an exceptional case, characteristic only of certain representatives of the order *Mycobacteria*. Thus, both AlkB1 and AlkB1A can be considered paralogs of AlkB0, suggesting functional divergence. Interestingly, it was AlkB0 that showed a discrepancy with the known signature motifs of alkane monooxygenases.

Thus, the clustering of the AlkB1 and AlkB2 types into one clade suggests evolutionary proximity and functional similarity, regardless of the presence of the *rubB* gene. This hypothesis is further supported by the relatively rare co-occurrence of AlkB1 and AlkB2 types within the same genome, indicating potential duplication of function. Furthermore, the near-ubiquitous presence of at least one of these types in almost all *Rhodococcus* (*sensu lato*) genomes suggests strong positive selection and, therefore, indicates their importance for bacterial metabolism.

Other AlkBs whose genes are localized without a neighboring rubredoxin gene form a separate, large, and diverse cluster. It is noteworthy that *Rhodococcoides corynebacterioides* and *Rhodococcoides kroppenstedtii*, which lack both AlkB1 and AlkB2, have an AlkB8 type that clusters with these two. We propose that the AlkB8 type evolved as a result of loss of co-localization with rubredoxin genes. The rapid divergence of AlkB8 and AlkB1-AlkB2 is consistent with the importance of rubredoxin genes in a genomic context.

Based on these results, we propose using the co-localization of *alkB* and *rubA* genes as a primary parameter for classifying rhodococcal AlkB family alkane monooxygenases. We suggest two generalized types: AlkBR (R for “rubredoxin”) and AlkBS (S for “single”) (Figure 6). It is important to note that studies on the biochemistry of alkane monooxygenases and their functional roles predominantly focus on AlkBR-type enzymes [54,66,88,89]. Thus, for all the intrinsic diversity of AlkBS-type alkane monooxygenases, our knowledge of their functionality is extremely limited, providing a rationale for focusing future research efforts on these enzymes.

## 4. Materials and Methods

### 4.1. Phylogenetic Analysis of Sequences of Alkane Monooxygenases of the AlkB Family

To obtain translated amino acid sequences of alkane monooxygenases of the AlkB family for representatives of the *Rhodococcus* (*sensu lato*) group, a blastp algorithm (blast.ncbi.nlm.nih.gov/Blast.cgi) was used to search for the sequence of alkane 1-monooxygenase *Rhodococcus rhodochrous* EP4 (AYA27164.1) in the GenBank database “non-redundant protein sequences (nr)” with a restriction on the taxa *Rhodococcus*, *Rhodococcoides*, *Prescottella*, and *Antrihabitans cavernicola* [90]. The obtained sequences were filtered by length (sequences shorter than 300 amino acids were removed), sequences marked as “partial” and “uncultured” were removed, and a few manually checked outliers were removed (Appendix A).

Sequence alignment was performed using the ClustalO v1.2.4 algorithm (ebi.ac.uk/Tools/msa/clustalo) [91]. The matrices of amino acid pairwise identity were obtained using the same tool.

The phylogenetic tree of amino acid sequences was constructed using the maximum likelihood method in the IQ-TREE v1.6.12 program (iqtree.cibiv.univie.ac.at) [92] with the search for the best substitution model by ModelFinder [93] and ultrafast bootstrap [94].

Visualization and annotation of phylogenetic trees were performed in the online service iTOL v7 (itol.embl.de) [95].

### 4.2. Assessment of the Taxonomy Classification

The taxonomic status of microorganisms was checked in the NCBI Taxonomy database [62], and the Genome Taxonomy Database (GTDB, release 220) (gtdb.ecogenomic.org) [63].

The taxonomic position of strains was clarified using the Type (Strain) Genome Server service (tygs.dsmz.de) [64].

The phylogenetic tree of the *Rhodococcus* (*sensu lato*) group and related taxa was constructed based on GenBank type strains and genomes using the GTDB-Tk v2.4.0 program without including GTDB reference genomes in multiple sequence alignment (“--skip_gtdb_refs” option for “align” command) [96].

### 4.3. Characterization of the Genomic Context

Genomic context was investigated using NCBI’s graphical GenBank and the online service SyntTax (archaea.i2bc.paris-saclay.fr/SyntTax) [97].

Gene clusters were searched using cblaster (cagecat.bioinformatics.nl/tools/search) [98].

Genomic regions were visualized using clinker (cagecat.bioinformatics.nl/tools/clinker) [99].

### 4.4. Characterization of Amino Acid Motifs

Motif discovery was performed in MEME Version 5.5.7 tool (meme-suite.org/meme/tools/meme) with parameters “One Occurrence Per Sequence (oops)” and motif width 6–15 [100].

Sequence logos were obtained from the WebLogo v2.8.2 online service (weblogo.berkeley.edu/logo.cgi).

## Figures and Tables

**Figure 1 ijms-26-01713-f001:**
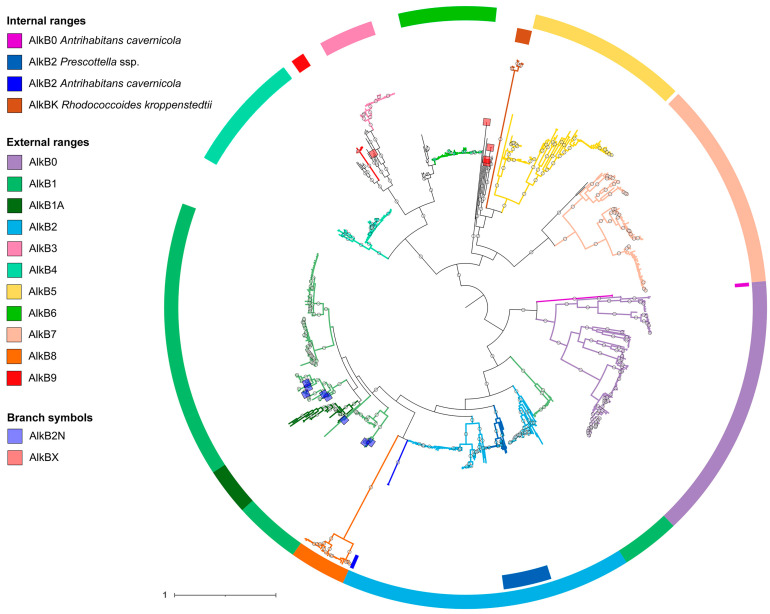
Midpoint rooted maximum likelihood phylogenetic tree of AlkB family alkane monooxygenases of the *Rhodococcus* (*sensu lato*) group. The tree was constructed using IQ-TREE; the automatically determined best-fit substitution model was JTT + F + I + G4. Branches with ultrafast bootstrap support values ≥ 95% (1000 replicates) are marked with white circles. The outer ranges and branch colors mark the proposed AlkB types. The inner ranges and branch colors mark the genus *Prescottella* ssp. (light blue), the species *Antrihabitans cavernicola* (blue, purple), and the AlkBK type of the species *Rhodococcoides kroppenstedtii*. The colored squares on the branches mark the AlkB2N (blue) and AlkBX (red) types.

**Figure 2 ijms-26-01713-f002:**
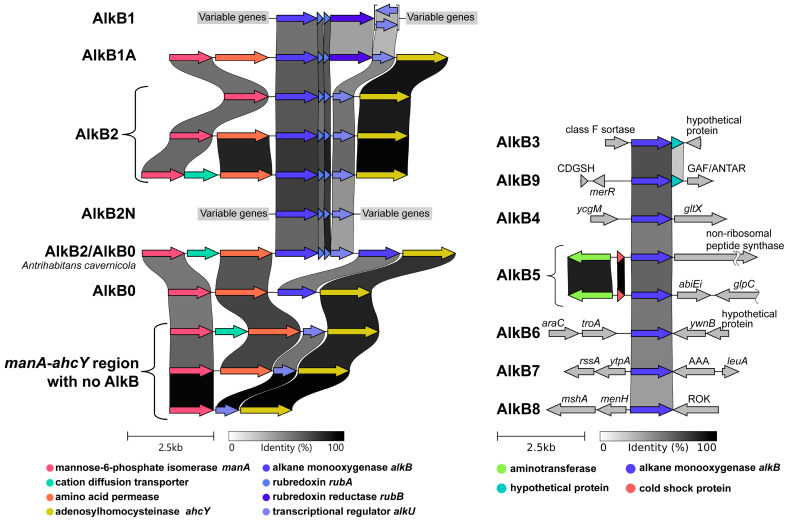
Composition of the conserved genomic regions for AlkB of different types. The arrows indicate corresponding genes, the links between entries indicate the percentage of identity of the corresponding amino acid sequences according to the “Identity” color bar. *merR*—MerR family transcriptional regulator; CDGSH—CDGSH iron–sulfur domain-containing protein; GAF/ANTAR—GAF and ANTAR domain-containing protein; *ycgM*—fumarylacetoacetate hydrolase family protein; *gltX*—glutamate–tRNA ligase; *abiEi*—AbiEi family antitoxin domain-containing protein; *glpC*—iron–sulfur-binding reductase; *araC*—AraC family transcriptional regulator; *troA*—TroA-like ABC transporter substrate-binding protein; *own*—NAD(P)H-binding protein; *rssA*—patatin-like phospholipase family protein; *ytpA*—phospholipase YtpA (alpha/beta hydrolase); AAA—ATP-binding protein (AAA + ATPase superfamily); *leuA*—2-isopropylmalate synthase; *mshA*—inositol-3-phosphate glycosyltransferase; *menH*—alpha/beta fold hydrolase; ROK—ROK family transcriptional regulator.

**Figure 3 ijms-26-01713-f003:**
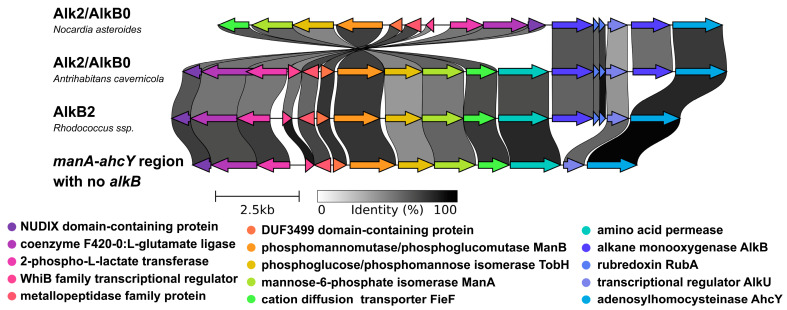
Comparison of different gene organization variants of the manA-ahcY region. The arrows indicate corresponding genes, the links between entries indicate the percentage of identity of the corresponding amino acid sequences according to the “Identity” color bar.

**Figure 4 ijms-26-01713-f004:**
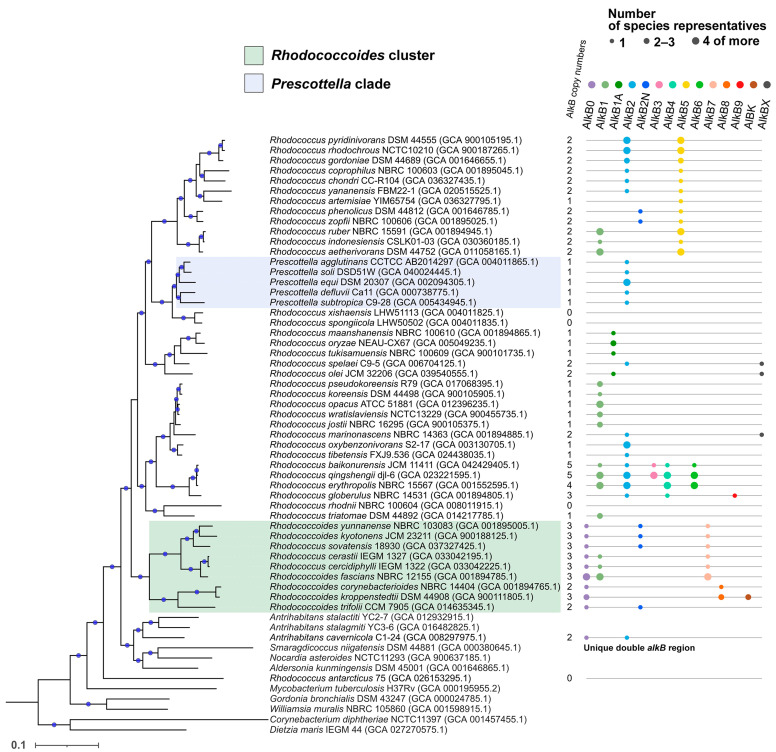
Midpoint rooted phylogenetic tree of type strains of the *Rhodococcus* (*sensu lato*) group and related species constructed using GTDB-tk. Branches with bootstrap support values ≥ 80% are marked with blue circles. The “AlkB copy numbers” column shows the number of AlkB types in the genomes of species representatives. The sizes of the labels correspond to the number of genomic regions considered in this paper for representatives of this species. Clusters containing all representatives of the genus *Rhodococcoides* and the clade consisting of all *Prescottella* ssp. are highlighted in color.

**Figure 5 ijms-26-01713-f005:**
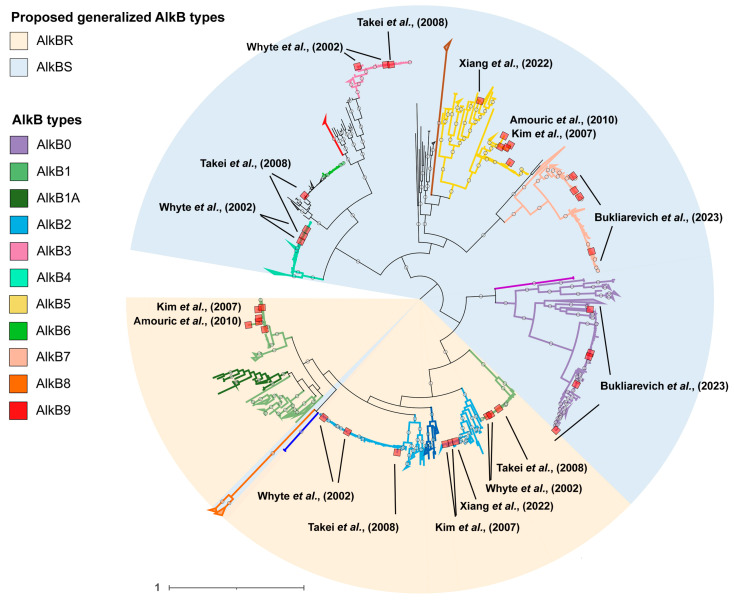
Midpoint rooted maximum likelihood phylogenetic tree of AlkB family alkane monooxygenases of the *Rhodococcus* (*sensu lato*) group with the addition of partial sequences of previously described alkane monooxygenases (according to Table 2). The tree was constructed using IQ-TREE; the automatically determined best-fit substitution model was JTT + F + I + G4. Branches with ultrafast bootstrap support values ≥ 95% (1000 replicates) are marked with white circles. The colored squares on the branches mark all previously described alkane monooxygenases (Table 2). The colored ranges correspond to the proposed generalized AlkB types [53,68,69,70,71,72].

**Figure 6 ijms-26-01713-f006:**
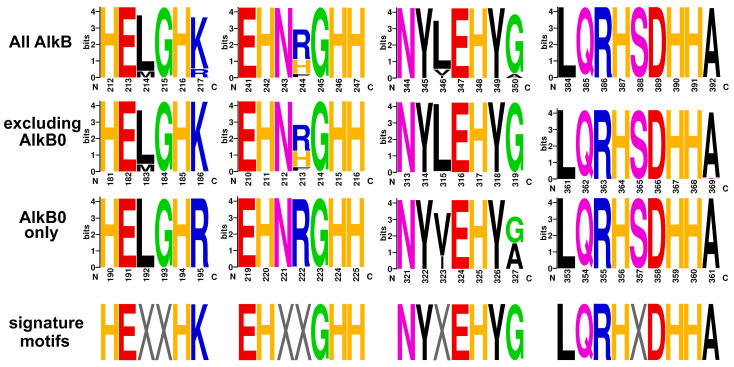
Sequence logos of different types of alkane monooxygenases for regions of signature amino acid motifs. Signature motifs are given according to [68,76].

**Table 1 ijms-26-01713-t001:** List of *Rhodococcus* species which have inconsistent taxonomic assignment in different databases.

NCBI Taxonomy	LPSN Status of NCBI Taxonomy	GTDB Taxonomy	TSGS Taxonomy
*Antrihabitans cavernicola*	correct name, synonym: *Rhodococcus cavernicola*	*Rhodococcus_E cavernicola*	*Rhodococcus cavernicola*
*Prescottella agglutinans*	correct name	*Rhodococcus agglutinans*	*Rhodococcus agglutinans*
*Prescottella defluvii*	correct name	*Rhodococcus defluvii*, *Rhodococcus defluvii_A*	*Rhodococcus defluvii*
*Prescottella equi*	correct name	*Rhodococcus equi*, *Rhodococcus equi_A*	*Rhodococcus equi*
*Prescottella soli*	synonym, correct name: *Rhodococcus soli*	absent	*Rhodococcus soli*
*Prescottella subtropica*	synonym, correct name: *Rhodococcus subtropicus*	*Rhodococcus subtropicus*	*Rhodococcus subtropicus*
*Rhodococcoides corynebacterioides*	synonym, correct name: *Rhodococcus corynebacterioides*	*Rhodococcus corynebacterioides*, *Rhodococcus corynebacterioides_A*	*Rhodococcus corynebacterioides*
*Rhodococcoides fascians*	synonym, correct name: *Rhodococcus fascians*	*Rhodococcus fascians*, *Rhodococcus fascians_E*	*Rhodococcus fascians*
*Rhodococcoides kroppenstedtii*	synonym, correct name: *Rhodococcus kroppenstedtii*	*Rhodococcus kroppenstedtii*	*Rhodococcus kroppenstedtii*
*Rhodococcoides kyotonense*	synonym, correct name: *Rhodococcus kyotonensis*	*Rhodococcus kyotonensis*, *Rhodococcus kyotonensis_B*	*Rhodococcus kyotonensis*
*Rhodococcoides trifolii*	synonym, correct name: *Rhodococcus trifolii*	*Rhodococcus trifolii*	*Rhodococcus trifolii*
*Rhodococcoides yunnanense*	synonym, correct name: *Rhodococcus yunnanensis*	*Rhodococcus yunnanensis*	*Rhodococcus yunnanensis*
*Rhodococcus antarcticus*	correct name	*Rhodococcus_D antarcticus*	*Rhodococcus antarcticus*
*Rhodococcus baikonurensis*	synonym, correct name: *Rhodococcus erythropolis*	*Rhodococcus qingshengii*	*Rhodococcus baikonurensis*
*Rhodococcus chondri*	correct name	absent	*Rhodococcus chondri*
*Rhodococcus indonesiensis*	correct name	*Rhodococcus* sp030360185	*Rhodococcus indonesiensis*
*Rhodococcus olei*	correct name	absent	*Rhodococcus olei*
*Rhodococcus qingshengii*	synonym, correct name: *Rhodococcus erythropolis*	*Rhodococcus qingshengii*, *Rhodococcus qingshengii_B*	*Rhodococcus qingshengii*
*Rhodococcus sovatensis*	correct name	absent	absent
*Rhodococcus tibetensis*	not validly published	*Rhodococcus* sp024438035	*Rhodococcus tibetensis*

**Table 2 ijms-26-01713-t002:** Previously described types of AlkB.

AlkB Type ^a^	Genomic Context If Available	AlkB GenBank ID	Strain	Source
AlkB1	RubA, RubA, RubB, AlkU	CAB51053.2	*R. erythropolis* NRRL B-16531	[68]
AlkB2	cationic transporter ^†^, RubA, RubA, AlkU ^†^	CAC37038.1		
AlkB3	(putative) exported protein ^†^	CAC40953.1		
AlkB4	glutamil t-RNA synthetase	CAC40954.1		
AlkB1	RubA, RubA, RubB, AlkU	AAK97448.1	*Rhodococcus* sp. Q15	
AlkB2	cationic transporter ^†^, RubA, RubA, AlkU	AAK97454.1		
AlkB3	—^b^	AAK97446.1		
AlkB4	—	AAK97447.1		
AlkB1 (AlkB7)	—	ABI13999.1 ^†^	*Rhodococcus* sp. DEE5151	[69]
AlkB1 (AlkB7)	—	ABI14001.1 ^†^	*Rhodococcus* sp. DEE5311	
AlkB1 (AlkB7)	—	ABI14004.1 ^†^	*Rhodococcus* sp. MOB100	
AlkB1 (AlkB7)	—	ABI14006.1 ^†^	*Rhodococcus* sp. THF100	
AlkB2 (AlkB6)	—	ABI14003.1 ^†^	*Rhodococcus* sp. DEOB100	
AlkB2 (AlkB6)	—	ABI14005.1 ^†^	*Rhodococcus* sp. MOP100	
AlkB5	—	ABI13998.1 ^†^	*R. rhodochrous* 116	
AlkB5	—	ABI14000.1 ^†^	*Rhodococcus* sp. DEE5301	
AlkB5	—	ABI14002.1 ^†^	*Rhodococcus* sp. DEE5316	
AlkB1	—	BAG06232.1 ^†^	*Rhodococcus* sp. TMP2	[70]
AlkB2	—	BAG06233.1 ^†^		
AlkB3	—	BAG06234.1 ^†^		
AlkB4	—	BAG06235.1 ^†^		
AlkB6 (AlkB5 ^c^)	—	BAG06236.1 ^†^		
AlkB1 (AlkB7)	RubA, RubA, RubB ^†^	ACX30747.1	*R. ruber* SP2B	[71]
AlkB5 (new or AlkB5)	—	ACX30751.1 ^†^		
AlkB1 (AlkB7)	—	ACX30755.1	*R. ruber* DSM 43338	
AlkB5 (new or AlkB5)	—	ACX30752.1 ^†^		
AlkB2	ManA, RubA, RubA, TetR, AhcY	WP_241385946.1	*Rhodococcus* sp. CH91	[72]
AlkB5 (new)	aminotransferase, cold-shock protein, AbiEi family protein	WP_241384812.1		
AlkB7	alpha/beta hydrolase	AMY23060	*R. fascians* PBTS 2	[53]
		AJW41446	*Rhodococcus* sp. B7740	
		QII08422	*R. fascians* A25f	
		QIH99153	*R. fascians* A2ld2	
AlkB0 (AlkB8)	ManA, AhcY	AYJ49258	*Rhodococcus* sp. PI Y	
		AMY51400	*R. fascians* D188	
		AMY24665	*R. fascians* PBTS 2	
		AJW39705	*Rhodococcus* sp. B7740	
		QII00755, QII06794 ^d^	*R. fascians* A2ld2, *R. fascians* A25f	

^a^ The designation used for the first time is given in parentheses if it differs from that proposed in the current work. ^b^ Not available. ^c^ Does not take into account the results of Kim et al. [69]. ^d^ Identical sequences. ^†^ Partial sequence.

## Data Availability

All data obtained in the work are included in the article and Appendix A.

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
