# Peer review of "Generalization of Classification of AlkB Family Alkane Monooxygenases from *Rhodococcus* (*sensu lato*) Group Based on Phylogenetic Analysis and Genomic Context Comparison"

_ijms, 2025, doi:10.3390/ijms26041713_

Round 1
Reviewer 1 Report
Comments and Suggestions for Authors
This is an interesting body of work that, in my opinion, does not yet fulfill its promise. Below I will detail both the major and minor concerns that the work raises for me. I am providing the comments in the order in which they appear in the text and am not distinguishing between what I consider major and minor comments but hope that they are clear from the context.
First some general comments:
As one reads the paper, it is not at all clear what the basis of the selected numbering system for the different categories is. Later it becomes clear that as much as possible the authors are trying to be consistent with prior work but that really is not apparent until near the end of the paper. I see a couple of approaches that could work. The introduction could begin by providing much more background on the rhodococcus specific classification scheme that has already been developed and then position this paper as a next step in that work. Alternatively perhaps the authors want to argue that an entirely new classification scheme is justified and star over.
It also is not clear that this work can be extended beyond Rhodococcus and if it can't be, how useful is it?
Figures 2 and 3 are confusing – why are some entries connected by a forward slash? The picture that corresponds to these combined entries doesn’t make sense – clearly the authors are trying to communicate something with those figures and esp with the decision to include some entries together but the meaning of that decision was not clear.
The work depends on the correct identification of the other genes in the genome context. Not at all sure that genome annotation is good enough for this to be reliable unless there is some other method in this paper to ascertain the reliability of all the annotations on which the classification schemes are based.
Significance of amino acid motifs is not clear – a lot has been written about defining histidine pattern. There are a few other amino acids now included in the motif identification but to simply list them without offering any idea about the functional significance is not interesting. There are now two published structures of AlkB and evidence that the AlphaFold3 structures are relevant so it is quite tractable for anyone interested in AlkB structures to thread sequences onto published structures or look at the AlphaFold3 structures and hypothesize about functional relevance of certain amino acids. In the absence of any discussion of the function of what the authors are describing as amino acid motifs, I can't appreciate the significance of including them.
There are some clear amino acid pattern differences between fatty acid desaturases and alkane hydroxylases. These differences have been published – page 15 – the implication here is that these amino acid sequences are the same for both protein families.
There is a lot of experimental evidence that other reductases can reduce AlkB rubredoxins – so necessity of having a dedicated reductase is unclear.
The idea that all AlkBs should be classified as either AlkBR and AlkBS for AlkBs that come with rubredoxins and those that do not, is interesting although in the absence of any functional significance to this idea, it is hard to know what to make of it.
Here begins the point by point comments that correspond to specific sections of the text. I hope it is clear from the amount of information provided about each comment precisely where in the text I am referring to.
1. The title is not justified by the work. It is not yet possible to say that this classification scheme is relevant for all bacterial genomes, which is what would be needed to be of significant use to the scientific community. The decision also to exclude AlkBs that are on plasmids greatly limits the scope of the work, although understandably make many parts of it easier.
2. There are a number of places where references are inserted but seem not to have been picked up by the reference management system so they appear not as superscripts but as field codes.
3. It is difficult to estimate the total amount of alkanes in the environment but if one accepts that argument about the cryptic hydrocarbon cycle, then the amount of alkanes from natural sources is considerably larger than the amount from anthropogenic sources.
4. This comment is part content/part writing: What is the evidence that copper containing enzymes oxidize substrates up to nine carbons in length? The wording of this entire paragraph is confusing – it makes it sound like all of these different enzymes are all copper enzymes, when they are clearly not. While sMMO can oxidize relatively large alkanes when pushed that is probably an artifact of the experimental set up and unlikely to be particularly significant to the global transformation of alkanes. This same paragraph seems to suggest that CYPs are the dominant enzymes for oxidation of medium chain length alkanes, which is not true. It is unclear precisely what the balance is in the environment between CYP and AlkB mediated oxidation of medium chain length alkanes but CYP is not the dominant enzyme.
5. “from C32 for AlmA”? Is that implying that it only oxidizes alkanes of chain lengths larger than 32 carbons? This is unclear. This paragraph should have a topic sentence. I would group those two enzymes together because they are the only two non-metalloenzymes among the enzymes that are important in the environment for oxidizing alkanes but the authors could have other intents – whatever the purpose of this paragraph and putting these two enzymes together is, it needs to be clear.
6. It is more correct to say that AlkB and the membrane spanning fatty acid desaturases are both members of the class-III diiron proteins, which also includes fatty acid hydroxylases, xylene monooxygenase, and UndB.
7. The three domain protein referenced in reference 49 is not a rubredoxin-rubredoxin reductase-AlkB fusion protein, rather it is a ferredoxin-ferredoxin reductase-AlkB fusion protein. The use of RubA and RubB is not particularly standard.
8. Desaturation is not that common with AlkB. It has been shown with norcarane but not with methyl phenyl cyclopropane or cyclohexane.
9. Page 5 – the amino acid limit for this work would not find AlkBs that have two and three domains.
10. Classification scheme – type A. Need to be more clear about what sort of rubredoxins are being considered. The first rubredoxin identified with AlkB (AlkG) is an unusual two domain rubredoxin where the C terminus is thought functional and the N terminus not, although each has an iron binding domain. However, there are many other cases where rubredoxins are associated with AlkB that have only the functional domain. In the OCT operon, there is another rubredoxin, AlkF, that has never been shown to be functional or shown to have any purpose. Is this the other rubredoxin that is being considered to be part of the type A category?
11. Need to make clear the functional significance of AlkU
12. Page 9 – The order of the introduction of the different categories is not clear - AlkB3, AlkB 4 then jump to AlkB6 – why?
13. It is not at all clear what defines AlkB9.
14. “the” non ribosomal peptide gene? There is more than one.
Comments on the Quality of English Language
1. Title while informative is not grammatical. One would have to say “allows ONE to propose generalized AlkB types” and that then sounds a bit too wordy.
2. Paragraph that starts on line 61 on page two is only a single sentence and could be incorporated into paragraph above without any loss of meaning
3. page 3 Line113 – some obvious typographical errors in this section.
4. Page 6 – make sure there are commas after every gene in these long lists – hard to understand without them.
Author Response
Thank you for your careful reading and detailed analysis of our paper, your comments have been very helpful. We have tried to take into account all your comments and have significantly revised some parts of the manuscript. Here are our point-by-point responses.
Reviewer 1
Comments: As one reads the paper, it is not at all clear what the basis of the selected numbering system for the different categories is. Later it becomes clear that as much as possible the authors are trying to be consistent with prior work but that really is not apparent until near the end of the paper. I see a couple of approaches that could work. The introduction could begin by providing much more background on the rhodococcus specific classification scheme that has already been developed and then position this paper as a next step in that work. Alternatively perhaps the authors want to argue that an entirely new classification scheme is justified and star over.
Response: We agree that the chosen presentation style can be confusing. We have revised and rewritten part of the Introduction address the concerns (lines 104-117) and have added a clarification to the Results section (line 174).
Comments: It also is not clear that this work can be extended beyond Rhodococcus and if it can't be, how useful is it?
Response: We believe that the approach used in this study can be applied to investigate the diversity of other genera of typical alkane-oxidizing microorganisms where AlkB-family alkane monooxygenases have been detected. Furthermore, our findings are relevant to the entire order Mycobacteriales, since we noted that the specificity of alkB gene localization is also observed in other genera within this order.
Comments: Figures 2 and 3 are confusing – why are some entries connected by a forward slash?
Response: Entries containing a forward slash, as well as all other transcript designations, correspond to the annotations provided in GenBank. For example, UKO88627.1 is annotated as "phosphomannomutase/phosphoglucomutase."
Comments: The picture that corresponds to these combined entries doesn’t make sense – clearly the authors are trying to communicate something with those figures and esp with the decision to include some entries together but the meaning of that decision was not clear.
Response: The links between entries indicate the percentage of identity of the corresponding amino acid sequences according to the “Identity” color bar. We have added this in the figure legends for Figures 2 and 3. The high sequence identity of proteins determines the conservation of a particular type region. We chose this visualization method because it also highlights gene rearrangements. For instance, in Figure 3, it illustrates the inversion of the upstream region, as well as the loss of the alkB0 gene and the alkB2 operon.
Comments: The work depends on the correct identification of the other genes in the genome context. Not at all sure that genome annotation is good enough for this to be reliable unless there is some other method in this paper to ascertain the reliability of all the annotations on which the classification schemes are based.
Response: While additional re-annotation of all sequences would improve accuracy, the key aspect for describing genomic context and confirming its conservation is the similarity of the corresponding proteins rather than their specific annotation. For example, in the AlkB6 type region, there is a hypothetical protein lacking a specific functional annotation. However, this gene is consistently present in all AlkB6-type regions, forming a conserved genomic context. Refining the annotation of this protein would not alter the overall genomic context of the AlkB6 type.
Comments: Significance of amino acid motifs is not clear – a lot has been written about defining histidine pattern. There are a few other amino acids now included in the motif identification but to simply list them without offering any idea about the functional significance is not interesting. There are now two published structures of AlkB and evidence that the AlphaFold3 structures are relevant so it is quite tractable for anyone interested in AlkB structures to thread sequences onto published structures or look at the AlphaFold3 structures and hypothesize about functional relevance of certain amino acids. In the absence of any discussion of the function of what the authors are describing as amino acid motifs, I can't appreciate the significance of including them.
There are some clear amino acid pattern differences between fatty acid desaturases and alkane hydroxylases. These differences have been published – page 15 – the implication here is that these amino acid sequences are the same for both protein families.
Response: Of course, the minor variations we identified cannot serve as definitive evidence of functional differences; rather, we aimed to highlight a hypothetical possibility. We have revised this part of Discussion section to better reflect the most significant points (lines 454-488). The primary objective of our study was to systematize, confirm, and generalize the results of previous research. We believe we have identified evidence suggesting that some widely accepted assumptions concerning the conserved motifs of alkane monooxygenases may require refinement.
Comments: There is a lot of experimental evidence that other reductases can reduce AlkB rubredoxins – so necessity of having a dedicated reductase is unclear.
Response: Indeed, and in our conclusion, we emphasize that colocalization with rubredoxin reductase should not be considered a key factor in the classification of AlkBs. However, we aimed to highlight the occurrence of such colocalization as an intriguing phenomenon that requires further investigation from both evolutionary and functional perspectives. We have rewritten this part of Discussion section (lines 492-515).
Comments: The idea that all AlkBs should be classified as either AlkBR and AlkBS for AlkBs that come with rubredoxins and those that do not, is interesting although in the absence of any functional significance to this idea, it is hard to know what to make of it.
Response: We have refined our conclusion to better convey our reasoning (lines 516-526). We intended to emphasize that experimental validation of alkane monooxygenase function has been performed only for the AlkBR type we propose. This is also very interesting from an evolutionary point of view, as alkB colocalization with rubredoxin genes is found across different domains but, on the other hand, appears to "disappear" within a single genus. In this article, we simply present these observations to draw attention to this phenomenon. However, we are currently conducting experimental research to explore this further.
Comments: 1. The title is not justified by the work. It is not yet possible to say that this classification scheme is relevant for all bacterial genomes, which is what would be needed to be of significant use to the scientific community. The decision also to exclude AlkBs that are on plasmids greatly limits the scope of the work, although understandably make many parts of it easier.
Response: We have revised the title to better reflect the scope and findings of the study.
Plasmids were not considered only when characterizing genomic regions and assessing the distribution of AlkB types across species, nor were incompletely assembled genomes. However, homologs found using BLAST included all AlkB sequences available in the GenBank database, including those encoded on plasmids. We have revised the manuscript to clarify this point and avoid ambiguity (lines 165-168).
Comments: 2. There are a number of places where references are inserted but seem not to have been picked up by the reference management system so they appear not as superscripts but as field codes.
Response: We have corrected the formatting of the references.
Comments: 3. It is difficult to estimate the total amount of alkanes in the environment but if one accepts that argument about the cryptic hydrocarbon cycle, then the amount of alkanes from natural sources is considerably larger than the amount from anthropogenic sources.
- This comment is part content/part writing: What is the evidence that copper containing enzymes oxidize substrates up to nine carbons in length? The wording of this entire paragraph is confusing – it makes it sound like all of these different enzymes are all copper enzymes, when they are clearly not. While sMMO can oxidize relatively large alkanes when pushed that is probably an artifact of the experimental set up and unlikely to be particularly significant to the global transformation of alkanes. This same paragraph seems to suggest that CYPs are the dominant enzymes for oxidation of medium chain length alkanes, which is not true. It is unclear precisely what the balance is in the environment between CYP and AlkB mediated oxidation of medium chain length alkanes but CYP is not the dominant enzyme.
- “from C32 for AlmA”? Is that implying that it only oxidizes alkanes of chain lengths larger than 32 carbons? This is unclear. This paragraph should have a topic sentence. I would group those two enzymes together because they are the only two non-metalloenzymes among the enzymes that are important in the environment for oxidizing alkanes but the authors could have other intents – whatever the purpose of this paragraph and putting these two enzymes together is, it needs to be clear.
Response: We have shortened the Introduction in accordance with Reviewer #2’s comments.
Comments: 6. It is more correct to say that AlkB and the membrane spanning fatty acid desaturases are both members of the class-III diiron proteins, which also includes fatty acid hydroxylases, xylene monooxygenase, and UndB.
Response: We have corrected this part of the Introduction (lines 58-61).
Comments: The three domain protein referenced in reference 49 is not a rubredoxin-rubredoxin reductase-AlkB fusion protein, rather it is a ferredoxin-ferredoxin reductase-AlkB fusion protein. The use of RubA and RubB is not particularly standard.
Response: Corrections regarding link to fusion protein have been made (lines 71-73).
Regarding the designations RubA and RubB, we have removed them from the Introduction (line 66). However, they have been retained in the rest of the text because they are commonly used in studies on Rhodococcus. The designations of rubredoxin as RubA and rubredoxin reductase as RubB were introduced in seminal papers: van Beilen et al. (2002, 10.1128/JB.184.6.1722-1732.2002), van Beilen et al. (2002, 10.1046/j.1462-2920.2002.00355.x), and Whyte et al. (2002, 10.1128/AEM.68.12.5933-5942.2002).
Since then, these designations have been used in many studies on rhodococci, e.g. Amouric et al. (2010, 10.1111/j.1365-2672.2009.04592.x), Xiang et al. (2023, 10.3390/microorganisms11061537), and Fenibo et al. (2023, 10.1016/j.scitotenv.2023.162951). Additionally, these designations are found in databases even for representatives of other taxa (e.g., UniProt: Q9HTK7, Q9HTK8, P42453, Q0VTB0).
This inconsistency in terminology reflects a broader issue of non-uniform nomenclature, which aligns with the lack of a unified classification of AlkB considered in our work.
Comments: 8. Desaturation is not that common with AlkB. It has been shown with norcarane but not with methyl phenyl cyclopropane or cyclohexane.
Response: We have revised the wording to clarify this point (line 75-77).
Comments: 9. Page 5 – the amino acid limit for this work would not find AlkBs that have two and three domains.
Response: Our filtering retained only sequences longer than 300 amino acids to remove partial proteins. This clarification has been added to the manuscript (line 533). Longer fusion proteins would have been present in the sample. However, they are not typical of rhodococci: in the raw BLAST data, only one such protein was found, and it was excluded from the final dataset as an outlier (Table S2).
Comments: 10. Classification scheme – type A. Need to be more clear about what sort of rubredoxins are being considered. The first rubredoxin identified with AlkB (AlkG) is an unusual two domain rubredoxin where the C terminus is thought functional and the N terminus not, although each has an iron binding domain. However, there are many other cases where rubredoxins are associated with AlkB that have only the functional domain. In the OCT operon, there is another rubredoxin, AlkF, that has never been shown to be functional or shown to have any purpose. Is this the other rubredoxin that is being considered to be part of the type A category?
Response: We did not focus on specific rubredoxin subtypes but rather on their colocalization with AlkB. The rubredoxins we found differ significantly from both AlkF and AlkG of the OCT operon: they contain a single domain and are notably shorter (approximately 50–60 amino acids). As a result, Rhodococcus rubredoxins are classified as a distinct type (van Beilen et al., 2002, 10.1128/JB.184.6.1722-1732.2002).
Comments: 11. Need to make clear the functional significance of AlkU
Response: To our knowledge, there are no studies devoted to the functional characterization of rhodococcal AlkU. Participation in regulation of alkB expression has been demonstrated only for homologs in other Mycobacteriales: Dietzia sp. (Liang et al., 2016, 10.1111/mmi.13232) and Mycobacterium tuberculosis (Stokas et al., 2022, 10.1128/spectrum.01969-22).
Comments: 12. Page 9 – The order of the introduction of the different categories is not clear - AlkB3, AlkB 4 then jump to AlkB6 – why?
Response: This order was chosen because types 3,4, and 6 cluster into a single clade, whereas AlkB5 forms a separate, highly diverse clade. However, we recognize that this ordering may not be intuitive, and we have revised this section for clarity (lines 277-286).
Comments: 13. It is not at all clear what defines AlkB9.
Response: AlkB9 proteins form a distinct and relatively distant clade, yet they exhibit high sequence similarity within this group, as reflected by their phylogenetic distances. Additionally, genes encoding AlkB9 share a conserved genomic context. At the same time, AlkB3 is found in Rhodococcus qingshengii and Rhodococcus erythropolis, AlkB9 is specific to Rhodococcus globerulus, which is good evidence of their isolation. We have clarified this point in the manuscript (lines 292-300).
Comments: 14. “the” non ribosomal peptide gene? There is more than one
Response: Corrections made (line 282).
Comments on the Quality of English Language
- Title while informative is not grammatical. One would have to say “allows ONE to propose generalized AlkB types” and that then sounds a bit too wordy.
Response: We have changed the title.
- Paragraph that starts on line 61 on page two is only a single sentence and could be incorporated into paragraph above without any loss of meaning
- page 3 Line113 – some obvious typographical errors in this section.
- Page 6 – make sure there are commas after every gene in these long lists – hard to understand without them.
Response: All necessary corrections made.
Reviewer 2
Comments: Basically, the authors re-analyzed the phylogenetic placements of AlkB gene sequences from Rhodococcus species, and proposed an improved classification of relevant clades. Although the methodologies are acceptable, the novelty of this work is minor, and organization of the manuscript has to be greatly revised.
Response: Thank you for this comment, indeed, the problem of refining the classification of AlkB was not reflected well enough by us in the manuscript. Previous studies on the detection of alkB types in rhodococci did not aim to establish a general classification. Phylogenetic analyses and genomic context comparisons were limited to individual alkB homologs. These works primarily described a few specific homologs in a sequential manner without any generalization was offered. We have revised the manuscript to highlight these issues and clarify our approaches to addressing them (lines 104-127).
Comments: The title:
The author stated that "a new phylogenetic approach" was applied. However, frankly, I did not get what new aspect was applied.
"huge diversity" is already known, rather than a new finding of this work.
Response: We have changed the title based on your comments and hope it better reflects the essence of the work.
Comments: Abstract:
"However, the reasons behind this redundancy and the functional diversity of the alkane monooxygenases encoded by these genes remain unclear." Despite this knowledge gap is identified by the authors, I do not think the authors have resolved this gap through the present work. How the classification was modified have to be clearly pointed out in the abstract, rather than just mentioned it. Which three novel types were proposed? The introduction is quite lengthy. Please put forward scientific questions rather than review the whole issue.
Response: The abstract and introduction have been revised to reflect the comments.
Comments: Lines 47, 145 and more places throughout the text: Only DOI was cited here.
Response: We have corrected the formatting of the references.

Reviewer 2 Report
Comments and Suggestions for Authors
Basically, the authors re-analyzed the phylogenetic placements of AlkB gene sequences from Rhodococcus species, and proposed an improved classification of relevant clades. Although the methodologies are acceptable, the novelty of this work is minor, and organization of the manuscript has to be greatly revised.
The title:
The author stated that "a new phylogenetic approach" was applied. However, frankly, I did not get what new aspect was applied.
"huge diversity" is already known, rather than a new finding of this work.
Abstract:
"However, the reasons behind this redundancy and the functional diversity of the alkane monooxygenases encoded by these genes remain unclear." Despite this knowledge gap is identified by the authors, I do not think the authors have resolved this gap through the present work. How the classification was modified have to be clearly pointed out in the abstract, rather than just mentioned it. Which three novel types were proposed? The introduction is quite lengthy. Please put forward scientific questions rather than review the whole issue. Lines 47, 145 and more places throughout the text: Only DOI was cited here. Comments on the Quality of English LanguageCan be improved.
Author Response
Comments: Basically, the authors re-analyzed the phylogenetic placements of AlkB gene sequences from Rhodococcus species, and proposed an improved classification of relevant clades. Although the methodologies are acceptable, the novelty of this work is minor, and organization of the manuscript has to be greatly revised.
Response: Thank you for this comment, indeed, the problem of refining the classification of AlkB was not reflected well enough by us in the manuscript. Previous studies on the detection of alkB types in rhodococci did not aim to establish a general classification. Phylogenetic analyses and genomic context comparisons were limited to individual alkB homologs. These works primarily described a few specific homologs in a sequential manner without any generalization was offered. We have revised the manuscript to highlight these issues and clarify our approaches to addressing them (lines 104-127).
Comments: The title:
The author stated that "a new phylogenetic approach" was applied. However, frankly, I did not get what new aspect was applied.
"huge diversity" is already known, rather than a new finding of this work.
Response: We have changed the title based on your comments and hope it better reflects the essence of the work.
Comments: Abstract:
"However, the reasons behind this redundancy and the functional diversity of the alkane monooxygenases encoded by these genes remain unclear." Despite this knowledge gap is identified by the authors, I do not think the authors have resolved this gap through the present work. How the classification was modified have to be clearly pointed out in the abstract, rather than just mentioned it. Which three novel types were proposed? The introduction is quite lengthy. Please put forward scientific questions rather than review the whole issue.
Response: The abstract and introduction have been revised to reflect the comments.
Comments: Lines 47, 145 and more places throughout the text: Only DOI was cited here.
Response: We have corrected the formatting of the references.

Round 2
Reviewer 1 Report
Comments and Suggestions for Authors
I still come away from reading the revised version of the paper unsure of what has been accomplished. The abstract is not helpful at all. It states "We hypothesize that alkane monooxygenases can be categorized based on the colocalization of alkB genes with rubredoxin genes.For the AlkB0 type, we identified deviations from previously reported conserved signature amino acid motifs." The value to the community of sorting AlkBs according to the gene context in which they are found, when the gene context relies on annotation, and the discussion of the gene context in this paper is not linked to any functional difference among these AlkBs is just not clear. It is still possible that with a major revision the paper could be streamlined so that the main points and its primary value to the community would be more clear. There are many efforts at genotyping AlkBs, starting with the key work of Nie et al in 2014. It seems that the authors downplay prior work. It just isn't clear why creating so many different categories based on nearby genes is important.
Comments on the Quality of English LanguageThere are numerous typographical and spelling errors in the manuscript. These will need to be fixed before the paper can be published.
Author Response
We sincerely appreciate the reviewer’s constructive comments, which have helped us identify and address key shortcomings in our manuscript.
In our view, a complete picture of the diversity of alkane monooxygenases in Rhodococcus is still lacking, and the existing approaches to classify them and to describe new types have notable limitations. In our work, we have tried to provide a general perspective on the problem of characterizing AlkBs diversity and to address these shortcomings.
The genomic context of alkB genes has been considered in previous studies on Rhodococcus, beginning with the early work of Whyte et al., 2002.
It is an important feature to distinguish between homologous AlkBs. Moreover, the variation in localization of the alkB gene as part of a functional operon and as a single entities is also of interest from an evolutionary and functional point of view. This is particularly relevant for the genus Rhodococcus known for harboring multiple homologs of alkane monooxygenases as it raises questions about the biological significance of the existence of such redundancy and about the differences in the biological processes in which these homologs participate.
We acknowledge that our study does not include functional characterization as well as a number of previous works devoted to the phylogenetic analysis and classification of alkane monooxygenases (10.1111/j.1462-2920.2007.01269.x 10.1038/srep04968 10.7717/peerj.14147). However, we strongly believe that constructing the most comprehensive picture of diversity of AlkB protein and its gene organization and elucidating evolutionary relationships will also advance the study of functional diversity.
In the revised manuscript, we have made substantial improvements, clarifying the gaps in the existing classification and emphasizing the importance of combining phylogenetic analysis with an assessment of genomic context as a feature of our approach. We hope these revisions address the reviewer’s concerns and improve the manuscript’s clarity.

Reviewer 2 Report
Comments and Suggestions for Authors
The authors have addressed my concerns.
Author Response
We have made significant changes to the manuscript in response to reviewer 1's comments. If you have any questions about the new version, we are ready to answer them.
